# OPTIMIZATION PLANNING FOR 3D CONVNETS

## ABSTRACT

3D Convolutional Neural Networks (3D ConvNets) have been regarded as a powerful class of models for video recognition. Nevertheless, it is not trivial to optimally learn a 3D ConvNets due to high complexity and various options of the training scheme. The most common hand-tuning process starts from learning 3D ConvNets using short video clips and then is followed by learning long-term temporal dependency using lengthy clips, while gradually decaying the learning rate from high to low as training progresses. The fact that such process comes along with several heuristic settings motivates the study to seek an optimal "path" to automate the entire training. In this paper, we decompose the path into a series of training "states" and specify the hyper-parameters, e.g., learning rate and the length of input clips, in each state. The estimation of the knee point on the performance-epoch curve triggers the transition from one state to another. We perform dynamic programming over all the candidate states to plan the optimal permutation of states, i.e., optimization path. Furthermore, we devise a new 3D ConvNets with a unique design of dual-head classifier to improve the spatial and temporal discrimination. Extensive experiments conducted on seven public video recognition benchmarks demonstrate the advantages of our proposal. With the optimization planning, our 3D ConvNets achieves superior results when comparing to the state-of-the-art video recognition approaches. More remarkably, we obtain the top-1 accuracy of 82.5% and 84.3% on the large-scale Kinetics-400 and Kinetics-600 datasets, respectively.

## 1 INTRODUCTION

The recent advances in 3D Convolutional Neural Networks (3D ConvNets) have successfully pushed the limits and improved the state-of-the-art of video recognition. For instance, an ensemble of LGD-3D networks (Qiu et al., 2019) achieves 17.88% in terms of average error in trimmed video classification task of ActivityNet Challenge 2019, which is dramatically lower than the error (29.3%) attained by the former I3D networks (Carreira & Zisserman, 2017). The result basically indicates the advantage and great potential of 3D ConvNets for improving the performance of video recognition. Despite these impressive progresses, learning effective 3D ConvNets for video recognition remains challenging, due to large variations and complexities of video content. Existing works on 3D ConvNets (Tran et al., 2015; Carreira & Zisserman, 2017; Tran et al., 2018; Wang et al., 2018c; Feichtenhofer et al., 2019; Qiu et al., 2017; 2019) predominately focus on the designs of network architectures but seldom explore how to train a 3D ConvNets in a principled way.

The difficulty in training 3D ConvNets originates from the high flexibility of the training scheme. Compared to the training of 2D ConvNets (Ge et al., 2019; Lang et al., 2019; Yaida, 2019), the involvement of temporal dimension in 3D ConvNets brings two new problems of *how many frames should be sampled from the video* and *how to sample these frames*. First, the length of video clip is a tradeoff to control the balance between training efficiency and long-range temporal modeling for learning 3D ConvNets. On one hand, training with short clips (16 frames) (Tran et al., 2015; Qiu et al., 2017) generally leads to fast convergence with large mini-batch, and also alleviates the overfitting problem through data augmentation brought by sampling short clips. On the other hand, recent works (Varol et al., 2018; Wang et al., 2018c; Qiu et al., 2019) have proven better ability in capturing long-range dependency when training with long clips (over 100 frames) at the expense of training time. The second issue is the sampling strategy. Uniform sampling (Fan et al., 2019; Jiang et al., 2019; Martínez et al., 2019) offers the network a fast-forward overview of the entire video,

while consecutive sampling (Tran et al., 2015; Qiu et al., 2017; 2019; Varol et al., 2018; Wang et al., 2018c) can better capture the spatio-temporal relation across frames. Given these complex choices of training scheme, learning a powerful 3D ConvNets often requires significant engineering efforts of human experts to determine the optimal strategy on each dataset. That motivates us to automate the design of training strategy for 3D ConvNets.

In the paper, we propose optimization planning mechanism which seeks the optimal training strategy of 3D ConvNets adaptively. To this end, our optimization planning studies three problems: *1) choose between consecutive or uniform sampling; 2) when to increase the length of input clip; 3) when to decrease the learning rate.* Specifically, we decompose the training process into several training states. Each state is assigned with the fixed hyper-parameters, including sampling strategy, length of input clip and learning rate. The transition between states represents the change of hyper-parameters during training. Therefore, the training process can be decided by the permutation of different states and the number of epochs for each state. Here, we build a candidate transition graph to define the valid transitions between states. The search of the best optimization strategy is then equivalent to seeking the optimal path from the initial state to the final state on the graph, which can be solved by dynamic programming algorithm. In order to determine the best epoch for each state in such process, we propose a knee point estimation method via fitting the performance-epoch curve. In general, our optimization planning is viewed as a training scheme controller and is readily applicable to train other neural networks in stages with multiple hyper-parameters.

To the best of our knowledge, our work is the first to address the issue of optimization planning for 3D ConvNets training. The issue also leads to the elegant view of how the order and epochs for different hyper-parameters should be planned adaptively. We uniquely formulate the problem as seeking an optimal training path and devise a new 3D ConvNets with dual-head classifier. Extensive experiments on seven datasets demonstrate the effectiveness of our proposal, and with optimization planning, our 3D ConvNets achieves superior results than several state-of-the-art techniques.

## 2 RELATED WORK

The early works using Convolutional Neural Networks for video recognition are mostly extended from 2D ConvNets for image classification (Karpathy et al., 2014; Simonyan & Zisserman, 2014; Feichtenhofer et al., 2016; Wang et al., 2016). These approaches often treat a video as a sequence of frames or optical flow images, and the pixel-level temporal evolution across consecutive frames are seldom explored. To alleviate this issue, 3D ConvNets in Ji et al. (2013) is devised to directly learn spatio-temporal representation from a short video clip via 3D convolution. Tran *et al.* design a widely-adopted 3D ConvNets in Tran et al. (2015), namely C3D, consisting of 3D convolutions and 3D poolings optimized on the large-scale Sports1M (Karpathy et al., 2014) dataset. Despite having encouraging performances, the training of 3D ConvNets is computationally expensive and the model size suffers from a massive growth. Later in Qiu et al. (2017); Tran et al. (2018); Xie et al. (2018), the decomposed 3D convolution is proposed to simulate one 3D convolution with one 2D spatial convolution plus one 1D temporal convolution. Recently, more advanced techniques are presented for 3D ConvNets, including inflating 2D convolutions (Carreira & Zisserman, 2017), non-local pooling (Wang et al., 2018c) and local-and-global diffusion (Qiu et al., 2019).

Our work expands the research horizons of 3D ConvNets and focuses on improving 3D ConvNets training by adaptively planning the optimization process. The related works for 2D ConvNets training (Chee & Toulis, 2018; Lang et al., 2019; Yaida, 2019) automate the training strategy via only changing the learning rate adaptively. Our problem is much more challenging especially when temporal dimension is additionally considered and involved in the training scheme of 3D ConvNets. For enhancing 3D ConvNets training, the recent works (Wang et al., 2018c; Qiu et al., 2019) first train 3D ConvNets with short input clips and then fine-tune the network with lengthy clips, which balances training efficiency and long-range temporal modeling. The multigrid method (Wu et al., 2020) further cyclically changes spatial resolution and temporal duration of input clips for a more efficient optimization of 3D ConvNets. The research in this paper contributes by studying not only training 3D ConvNets with multiple lengths of input clips, but also adaptively scheduling the change of input clip length through optimization planning.

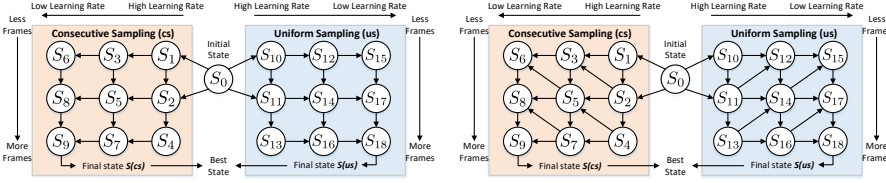

| (a) Basic transition graph | (b) Extended transition graph |

Figure 1: Examples of two transition graphs. The circles denote candidate states and the arrows represent candidate transitions. The ultimate model is the one with higher accuracy of the two final states.

## 3 OPTIMIZATION PLANNING

### 3.1 PROBLEM FORMULATION

The goal of optimization planning is to automate the learning strategy of 3D ConvNets. Formally, the optimization process of 3D ConvNets can be represented as an optimization path $\mathcal{P} = \langle S_0, S_1, ..., S_N \rangle$, which consists of one initial state $S_0$ and $N$ intermediate states. Each intermediate state is assigned with the fixed hyper-parameters, and the training is performed with these $N$ different settings one by one. The training epoch on each setting is decided by $\mathcal{T} = \{t_1, t_2, ..., t_N\}$, in which $t_i$ denotes the number of epochs when moving from $S_{i-1}$ to $S_i$. The hyper-parameters include $sampling\_strategy \in \{cs, us\}$, $length\_of\_input\_clip \in \{l_1, l_2, ..., l_{N_l}\}$ and $learning\_rate \in \{r_1, r_2, ..., r_{N_r}\}$, where $cs$ and $us$ denotes consecutive sampling and uniform sampling, respectively. In this case, there are $2 \times N_l \times N_r$ valid types of training states.

The objective function of optimization planning is to seek the optimal strategy $\{\mathcal{P}, \mathcal{T}\}$ by maximizing the performance of the final state $S_N$:

$$\underset{\mathcal{P}, \mathcal{T}}{maximize} \, \mathcal{V}(S_N), \qquad (1)$$

where $\mathcal{V}(\cdot)$ is the target performance, i.e., mean accuracy on validation set in our case.

### 3.2 OPTIMIZATION PATH

To plan the optimal permutation of training states, we first choose a final state $S_N$, which is usually with low learning rate and lengthy input clip. Then, the problem of seeking an optimal optimization path to $S_N$ is naturally decomposed to the subproblem of finding the optimization path to an intermediate state $S_i$ and the state transition from $S_i$ to $S_N$. As such, the problem can be solved by dynamic programming. Specifically, the solution of optimization path $\mathcal{P}(S_N)$ can be given in a recursive form:

$$\mathcal{P}(S_N) = \langle \mathcal{P}(S_{i^*}), S_N \rangle, \quad i^* = \underset{i}{argmax} \left\{ \mathcal{V}(S_i \to S_N) \right\}. \qquad (2)$$

When executing the transfer from the state $S_i$ to the state $S_N$, we fine-tune the 3D ConvNets at the state $S_i$ by using the hyper-parameters at the state $S_N$. We then evaluate such fine-tuned model on the validation set to measure the priority of this transition, i.e., $\mathcal{V}(S_i \to S_N)$. We choose the state $S_i^*$, which achieves the highest priority of transition to the state $S_N$, as the preceding state of $S_N$. In other words, the optimal path for $S_N$ derives from the best-performing preceding state $S_{i^*}$. Here, we propose to pre-define all the valid transitions in a directed acyclic graph and determine the best optimization path of each state one by one in the topological order. Figure 1(a) shows one example of the pre-defined transition graph. In the example, we set the number of candidate input clip lengths $N_l = 3$ and the number of candidate learning rates $N_r = 3$. Hence, there are $2 \times 3 \times 3 = 18$ candidate states. Then, the possible transitions, i.e., the connections between states, are determined by the following principles:

(1) The transitions between states with different sampling strategies are forbidden. We choose $S_9$ and $S_{18}$ as the final states for consecutive sampling and uniform sampling, respectively.

(2) The training only starts from high learning rate and short input clips.

(3) The intermediate state can be only transferred to a new state, where either the learning rate is decreased or the length of input clip is increased in the new state.

Please note that, some very specific learning rate strategies, e.g., schedules with restart or warmup, show that increasing the learning rate properly may benefit training. Nevertheless, there is still no

Table 1: The comparisons of four fitting functions in terms of RMSE and R-Square.

| Fitting Function $f_\alpha(t)$ | Constraints | RMSE | R-Square |
|---|---|---|---|
| **power:** $\alpha_1 + \alpha_2(t+1)^{\alpha_3} + \alpha_4 t + \alpha_5 t^2$ | $\alpha_2, \alpha_3, \alpha_5 < 0$ | $1.010 \times 10^{-3}$ | 0.356 |
| **multi-power:** $\alpha_1 + \alpha_2(t+1)^{\alpha_3} + \alpha_4(t+1)^{\alpha_5} + \alpha_6 t + \alpha_7 t^2$ | $\alpha_2, \alpha_3, \alpha_4, \alpha_5, \alpha_7 < 0$ | $1.030 \times 10^{-3}$ | 0.320 |
| **exponential:** $\alpha_1 + \alpha_2 e^{\alpha_3 t} + \alpha_4 t + \alpha_5 t^2$ | $\alpha_2, \alpha_3, \alpha_5 < 0$ | $\mathbf{1.007 \times 10^{-3}}$ | **0.360** |
| **multi-exponential:** $\alpha_1 + \alpha_2 e^{\alpha_3 t} + \alpha_4 e^{\alpha_5 t} + \alpha_6 t + \alpha_7 t^2$ | $\alpha_2, \alpha_3, \alpha_4, \alpha_5, \alpha_7 < 0$ | $1.063 \times 10^{-3}$ | 0.350 |

Figure 2: The examples of (a) the collected performance-epoch curves; (b) the fitting results for training model from scratch; (c) the fitting results for fine-tuning model.

clear principle of when to increase the learning rate, and thus it is very difficult to automate these schedules. In the works of adaptively changing the learning rate for 2D ConvNets training (Ge et al., 2019; Lang et al., 2019; Yaida, 2019), such cyclic schedules are also not taken into account. As a result, we only consider the schedule of decreasing learning rate in the transition graph.

These principles can simplify the transition graph and reduce the time cost when solving Equ.(2). We take this graph as **basic transition graph**. Furthermore, we also build an **extended transition graph** by enabling simultaneously decreasing the input clip length and the learning rate, as shown in Figure 1(b). In such graph, the training strategies are more flexible.

### 3.3 STATE TRANSITION

One state transition from $S_i$ to $S_j$ is defined as a training step that starts to optimize the model at $S_i$ by using the hyper-parameters at $S_j$. Then the question is when this training step completes. Here, we derive the spirit from SASA (Lang et al., 2019) that trains the network with constant hyper-parameters until it reaches a stationary condition. SASA presents to adaptively evaluate the convergence of stochastic gradient descent by Yaida's condition (Yaida, 2019) during training. However, in practice, the thoroughly optimized network does not always perform well on validation set due to overfitting problem. Therefore, we take both convergence and overfitting into account, and propose to estimate the knee point on the performance-epoch curve evaluated on the validation set, which performs more steadily across various datasets. Specifically, we measure the accuracy $y_t$ by evaluating the intermediate model after $t$-th training epoch on validation set. To estimate the knee point given a limited number of observations $y_t$, we fit the curve by a continuous function $f_\alpha(t)$ as

$$y_t = f_\alpha(t) + z_t, \ \ z_t \sim \mathcal{N}(0, \sigma^2), \tag{3}$$

where $z_t$ is the stochastic factor following a normal distribution, and $\alpha$ denotes the parameters of function $f$. Here, we choose $f_\alpha(t)$ as a unimodal function to ensure that there is only one maximum value. The curve fitting can be formulated as the optimization of parameters $\alpha$ by minimizing the distance between observed performance and estimated performance:

$$\underset{\alpha}{minimize} \sum_0^t \|y_t - f_\alpha(t)\|^2, \ \ \ s.t. \ f_\alpha(t) \text{ is unimodal.} \tag{4}$$

We exploit Trust Region Reflective algorithm (Branch et al., 1999) to solve this problem and the algorithm is robust for arbitrary form of function $f_\alpha(t)$. To adaptively stop the iteration, we estimate the knee point epoch $t^*$ by solving Equ.(4) after each training epoch. If the current epoch $t$ is larger than $t^* + T$, we will stop the iteration and choose $t^*$ as the best epoch number. Here, $T$ is a delay parameter which allows the model to have a $T$-epoch attempt even if $t > t^*$. We simply fix the delay parameter $T$ to 10 in all the experiments.

Next, the essential issue is the form of fitting function $f_\alpha(t)$. We separate the function into two parts $f_\alpha(t) = g_\alpha(t) + h_\alpha(t)$, where $g_\alpha(t)$ is an increasing bounded function to simulate the convergence

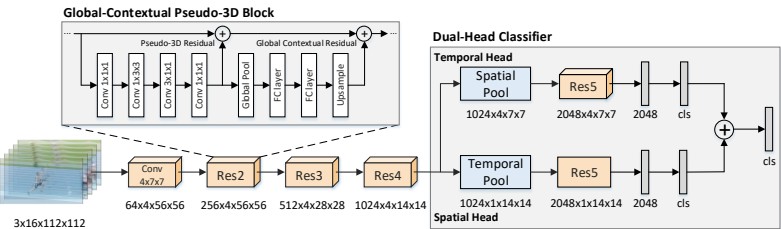

Figure 3: An overview of our proposed Dual-head Global-contextual Pseudo-3D (DG-P3D) network. Here, we take the 16-frame input as an example and the size of output feature map is also given for each layer.

of the model, and $h_\alpha(t)$ is a concave function to model the influence of overfitting. Table 1 shows four examples of fitting function $f_\alpha(t)$. In the four functions, we fix $h_\alpha(t)$ as a quadratic function and choose **power**, **multi-power**, **exponential** and **multi-exponential** function as $g_\alpha(t)$, respectively. Please note that, for each function, some constraints are given to guarantee the properties of $g_\alpha(t)$ and $h_\alpha(t)$. We empirically validate the functions by pre-collecting 162 performance-epoch curves (Figure 2(a)) from the training processes of different networks on different datasets and employing the four functions to fit the curves by solving Equ.(4). Table 1 compares the average Root Mean Square Error (**RMSE**) and **R-square** when using each function. Figure 2(b) and Figure 2(c) further depict a fitting example in the context of model training from scratch and model fine-tuning, respectively. The general observation is that, all the four functions can nicely fit the performance-epoch curve and do not make a major difference on the final performance. Thus, we simply choose the best-performing exponential function in the rest of the paper.

## 4 3D CONVNETS ARCHITECTURE

In this section, we present the proposed **Dual-head Global-contextual Pseudo-3D** (DG-P3D) network. An overview of the architecture is shown in Figure 3. In particular, the network is originated from the residual network (He et al., 2016) and further extended to 3D manner with three designs, i.e., pseudo-3D convolution, global context and dual-head classifier.

**Pseudo-3D convolution.** To achieve a good tradeoff between accuracy and computational cost, pseudo-3D convolution is proposed in Qiu et al. (2017) that decomposes 3D learning into 2D convolutions in spatial space plus 1D operations in temporal dimension. The similar idea of decomposing 3D convolution is also presented in R(2+1)D (Tran et al., 2018) and S3D (Xie et al., 2018). To simplify the decomposition, in this paper, we only choose P3D-A block with the highest performance in Qiu et al. (2017), which cascades the spatial convolution and temporal convolution in turn.

**Global context.** The recent works on non-local networks (Wang et al., 2018c; Cao et al., 2019; Qiu et al., 2019) highlight the drawback of performing convolutions, in which each operation processes only a local window of neighboring pixels and lacks a holistic view of field. To address this limitation, we choose the simple way to encapsulate global context that learns the global residual from the global-pooled representation and then broadcasts to each position in the feature map.

**Dual-head classifier.** 3D ConvNets are expected to have both spatial and temporal discrimination. For example, the SlowFast network (Feichtenhofer et al., 2019) contains one separate pathway for visual appearance and temporal dynamics, respectively. Here, we uniquely propose a simpler way that builds a dual-head classifier at the top of the network instead of the two-path structure in the SlowFast network. In between, the temporal head with large temporal dimension focuses on modeling the temporal evolution, and the spatial head with large spatial resolution emphasizes the spatial discrimination. The predictions from two heads are linearly fused. As such, our design costs less computations and is easier to implement.

## 5 EXPERIMENTS

### 5.1 DATASETS

The experiments are conducted on HMDB51, UCF101, ActivityNet, SS-V1/V2, Kinetics-400 and Kinetics-600 datasets. Table 2 details the information and settings on these datasets. The HMDB51

Table 2: The number of videos, the number of target categories and the detailed settings for optimization planning on HMDB51, UCF101, ActivityNet, SS-V1, SS-V2, Kinetics-400 and Kinetics-600 datasets.

| Dataset | #videos | #classes | $l_1$ | $l_2$ | $l_3$ | $r_1$ | $r_2$ | $r_3$ | Dropout |
|---|---|---|---|---|---|---|---|---|---|
| HMDB51 | 6K | 51 | 16 | 32 | 64 | 0.01 | 0.001 | 0.0001 | 0.9 |
| UCF101 | 13K | 101 | 16 | 32 | 64 | 0.01 | 0.001 | 0.0001 | 0.9 |
| ActivityNet | 20K | 200 | 16 | 32 | 128 | 0.01 | 0.001 | 0.0001 | 0.9 |
| SS-V1 | 108K | 174 | 16 | 32 | – | 0.04 | 0.004 | 0.0004 | 0.5 |
| SS-V2 | 220K | 174 | 16 | 32 | – | 0.04 | 0.004 | 0.0004 | 0.5 |
| Kinetics-400 | 300K | 400 | 16 | 32 | 128 | 0.04 | 0.004 | 0.0004 | 0.5 |
| Kinetics-600 | 480K | 600 | 16 | 32 | 128 | 0.04 | 0.004 | 0.0004 | 0.5 |

(Kuehne et al., 2011), UCF101 (Soomro et al., 2012), Kinetics-400 (Carreira & Zisserman, 2017) and Kinetics-600 (Carreira et al., 2018) are the most popular video benchmarks for action recognition on trimmed video clips. The Something-Something V1 (SS-V1) dataset is firstly constructed in Goyal et al. (2017) to learn fine-grained human-object interactions, and then extended to Something-Something V2 (SS-V2) recently. The ActivityNet (Caba Heilbron et al., 2015) dataset is an untrimmed video benchmark for activity recognition. The latest released version of the dataset (v1.3) is exploited. In our experiments, we only use the video-level label of ActivityNet and disable the temporal annotations. Note that the labels for test sets are not publicly available, and thus the performances of ActivityNet, SS-V1, SS-V2, Kinetics-400 and Kinetics-600 are all reported on the validation set. For optimization planning, the original training set of each dataset is split into two parts for learning the network weights and validating the performance, respectively. We construct this internal validation set with the same size as the original validation/test set. Note that the original validation/test set is never exploited in the optimization planning.

## 5.2 IMPLEMENTATION DETAILS

For **optimization planning**, we set the number of choices for both input clip length $N_l$ and learning rate $N_r$ as 3, and utilize the extended transition graph introduced in Section 3.2. The candidate values of input clip length $\{l_1, l_2, l_3\}$ and learning rate $\{r_1, r_2, r_3\}$ for each dataset are summarized in Table 2. Specifically, on SS-V1, SS-V2, Kinetics-400 and Kinetics-600 datasets, the base learning rate is set as $0.04$ and the dropout ratio is fixed as $0.5$. For HMDB51, UCF101 and ActivityNet, we set lower base learning rate and higher dropout ratio due to limited training samples. The maximum clip length is 64 for HMDB51 and UCF101, while increased to 128 for ActivityNet, Kinetics-400 and Kinetics-600. Considering that the video clips in SS-V1 and SS-V2 are usually shorter than 64 frames, we only use two settings, i.e., 16-frame and 32-frame, for the input clip.

The **network training** in this paper is implemented on Caffe (Jia et al., 2014) framework and the mini-batch stochastic gradient descent is employed to tune the network. The resolution of the input clip is fixed as $224 \times 224$, which is randomly cropped from the video clip resized with the short size in $[256, 340]$. The clip is randomly flipped along horizontal direction for data augmentation except for SS-V1 and SS-V2 in view of the direction-related categories. Following the settings in Wang et al. (2018c); Qiu et al. (2019), for the network training with long clips (64-frame and 128-frame), we freeze the parameters of all Batch Normalization layers except for the first one since the batch size is too small for batch normalization.

There are two **inference strategies** for the evaluations. The first one roughly predicts the video label on a $224 \times 224$ single center crop from the centric one clip resized with the short size 256. This strategy is only used when planning the optimization for the purpose of efficiency. Once the optimization path is fixed, we train 3D ConvNets with the path and evaluate the learnt 3D ConvNets by using the second strategy, i.e., the three-crop strategy as in Feichtenhofer et al. (2019), which crops three $256 \times 256$ regions from each video clip. The video-level prediction score is achieved by averaging all scores from 10 uniform sampled clips.

## 5.3 EVALUATION OF OPTIMIZATION PLANNING

We firstly verify the effectiveness of our proposed optimization planning for 3D ConvNets and compare the hand-tuned strategies. To find the most powerful hand-tuned strategy, we capitalize on the popular practices in the literature, and grid-search the training settings through four dimensions, i.e., input length, learning rate decay, sampling strategy and training epochs. Specifically, for input clip

Table 3: The comparisons between optimization planning (OP) and hand-tuned strategies with different 3D ConvNets on Kinetics-400 dataset. The number in the bracket denotes the best number of epoches, which is achieved by grid-search for hand-tuned strategies and adaptively determined for our optimization planning.

| Network | Sampling | Hand-tuned Strategies | | | | | | OP |
| | | $l_1 \rightarrow l_3$ | | $l_2 \rightarrow l_3$ | | $l_1 \rightarrow l_2 \rightarrow l_3$ | | |
| | | 3-step | cosine | 3-step | cosine | 3-step | cosine | |
|---|---|---|---|---|---|---|---|---|
| P3D | *consecutive* | 74.5 (256) | 74.4 (320) | **75.4** (320) | 75.3 (384) | 75.0 (320) | 75.2 (256) | **76.8** (184) |
| | *uniform* | 73.9 (192) | 74.3 (256) | 74.7 (256) | 75.0 (192) | 74.9 (256) | 75.2 (192) | |
| G-P3D | *consecutive* | 75.4 (384) | 75.7 (256) | 76.0 (320) | 76.0 (256) | **76.2** (320) | 75.7 (320) | **77.1** (234) |
| | *uniform* | 75.0 (256) | 75.5 (192) | 75.5 (192) | 75.8 (256) | 75.4 (320) | 75.9 (192) | |
| DG-P3D | *consecutive* | 76.9 (256) | 77.0 (192) | 77.3 (320) | **77.4** (256) | 77.0 (320) | 77.3 (256) | **78.3** (219) |
| | *uniform* | 76.1 (192) | 76.3 (128) | 76.3 (256) | 76.5 (256) | 76.2 (192) | 76.5 (192) | |

Table 4: The comparisons between optimization planning and hand-tuned strategy with DG-P3D on HMDB51 (split 1), UCF101 (split1), ActivityNet, SS-V1, SS-V2 and Kinetics-400 datasets. The backbone is ResNet-50 pre-trained on ImageNet. The time cost for grid search/optimization planning is reported with 8 NVidia Titan V GPUs in parallel.

| Strategy | | HMDB51 | UCF101 | ActivityNet | SS-V1 | SS-V2 | Kinetics-400 |
|---|---|---|---|---|---|---|---|
| Hand-tuned | *top-1* | 53.7 | 86.2 | 74.2 | 51.0 | 62.9 | 77.4 |
| Strategy | *time cost* | 83h | 158h | 166h | 540h | 1072h | 4057h |
| Optimization | *top-1* | **55.4** | **87.4** | **76.5** | **51.8** | **64.5** | **78.3** |
| Planning | *time cost* | 6h | 13h | 38h | 67h | 142h | 288h |

length, we follow the common training scheme that first learns the network with short clips and then fine-tunes the network on lengthy clips, and experiment with three strategies $l_1 \rightarrow l_3$, $l_2 \rightarrow l_3$ and $l_1 \rightarrow l_2 \rightarrow l_3$. For each input clip length, we train the network with the same number of epochs. For learning rate decay, we choose two mostly utilized strategies, i.e., 3-step learning rate decay (Wang et al., 2018c; Qiu et al., 2019) and cosine decay (Feichtenhofer et al., 2019). The optimal training epoch for each strategy is determined by grid-searching from $[128, 192, 256, 320, 384]$ epochs.

Table 3 shows the comparisons between optimization planning and hand-tuned strategies with three architectures P3D, G-P3D and DG-P3D on Kinetics-400 dataset. All the networks are derived from the ResNet-50 pre-trained on ImageNet dataset. The **P3D** network extends the original ResNet-50 to 3D network by utilizing pseudo-3D convolutions. **G-P3D** and **DG-P3D** further employ the global context and global context plus dual-head classifier, respectively. Overall, the proposed optimization planning consistently leads to a performance boost against the best hand-tuning strategy on three networks by 1.4%, 0.9% and 0.9%, respectively. The results basically indicate the advantage of dynamically determining the training strategy. Although the number of epochs for each hand-tuned strategy is tuned by grid-search, the strategy of optimization planning is more flexible and exhibits higher performance. Moreover, with the same optimization planning strategy, DG-P3D network achieves 1.2% improvement over G-P3D, which validates the proposed dual-head classifier.

Taking our DG-P3D as 3D ConvNets, Table 4 details the comparisons between optimization planning and the hand-tuned strategy across six different datasets. The accuracy of the hand-tuned strategy is reported on the best training scheme by grid search on each dataset. Such best hand-tuned strategy can be considered as a well-tuned DG-P3D model without optimization planning. The time cost of optimization planning contains the training time of exploring all the possible transitions, and that of hand-tuned strategy is measured by grid-searching the candidate training strategies. Compared to the hand-tuned strategy, optimization planning shows consistent improvements across different datasets, and requires much less time than the exhaustive grid search due to adaptive determination of training scheme. Figure 4 further depicts the optimal optimization paths on different datasets. An interesting observation is that SS-V1/2 tend to select uniform sampling while Kinetics-400 prefers consecutive sampling. We speculate that this may be the result of different emphases of the two sampling strategies. In general, the most special on uniform sampling is to capture the completeness of a video with only a small number of sampled frames. In contrast, consecutive sampling emphasizes the continuity in a video but may only focus on a part of the video content. The SS-V1/V2 datasets consist of fine-grained interactions and the differentiation between these interactions relies more on the completeness of an action. For example, it is almost impossible to distinguish the videos of the category "Pushing something so that it falls off the table" from those of "Pushing something so that it almost falls off but doesn't," if only based on part of the video content. In other words, uniform sampling offers the completeness of a video and benefits the recognition on

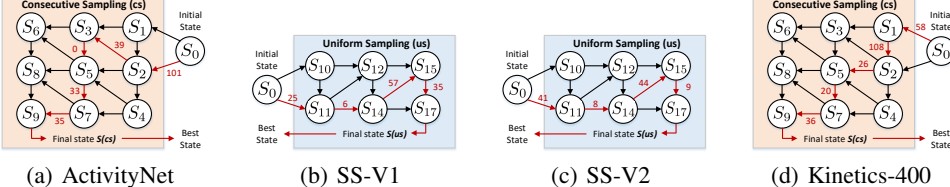

| (a) ActivityNet | (b) SS-V1 | (c) SS-V2 | (d) Kinetics-400 |

Figure 4: The best optimization path produced by the proposed optimization planning on (a) ActivityNet; (b) SS-V1; (c) SS-V2; (d) Kinetics-400. The red edge represents the state transition in the optimization path, while the black edges denote the transitions that have been explored but not selected in the final optimization path. The optimal number of training epochs is also given for each transition in the path.

SS-V1/V2. Instead, the videos in Kinetics-400 are usually with static scenes or slow motion. Hence, the completeness may not be essential in this case, but consecutive sampling encodes the continuous changes across frames and thus captures the spatio-temporal relation better.

## 5.4 MORE EXPERIMENTAL ANALYSIS ON OPTIMIZATION PLANNING

Next, we analyze the impact of our optimization planning from two more perspectives: 1) performance difference using different optimization paths, and 2) the transfer of optimization path across different 3D ConvNets. With regard to the former aspect, we experiment with some variant paths on UCF101, which are built by either inserting an additional state or skipping an intermediate state in our adopted optimization path. For fair comparisons, the numbers of epochs in these variant paths are re-determined by the algorithm in Section 3.3. The results indicate that inserting and skipping one state result in an accuracy decrease of 0.2%~1.0% and 0.3%~1.5%, respectively. For the latter one, we conduct the experiments by utilizing the optimal path found with DG-P3D on Kinetics-400 as the path for I3D (Carreira & Zisserman, 2017). Training I3D with such optimization path achieves the accuracy of 73.8% on Kinetics-400 with RGB input and leads to 1.7% performance improvement against the original I3D model. The results again demonstrate the effectiveness of our optimization planning and basically validate the generalizability of the learnt strategy across different networks.

## 5.5 COMPARISONS WITH STATE-OF-THE-ART

We compare with several state-of-the-art techniques on **HMDB51**, **UCF101** and **ActivityNet** datasets. The performance comparisons are summarized in Table 5. The backbone of DG-P3D is either ResNet-50 or ResNet-101 pre-trained on ImageNet. Please note that most recent works employ Kinetics-400 pre-training to improve the accuracy. Here, we also choose the two-step strategy that first trains DG-P3D on Kinetics-400 (K400) and then fine-tunes the network on the target dataset. The two steps are both trained with optimization planning. Overall, DG-P3D achieves the highest performances on all the three datasets, i.e., 78.8% on HMDB51, 97.8% on UCF101 and 86.8% on ActivityNet. In particular, DG-P3D outperforms the other 3D ConvNets of I3D, R(2+1)D, S3D-G and LGD-3D by 4.3%, 4.3%, 2.9% and 3.1% on HMDB51, respectively. The results again verify the merit of the learnt 3D ConvNets. For ActivityNet, most baselines utilize the temporal annotation to locate the foreground segment in the untrimmed videos. In our experiments, we only use the video-level annotations and our DG-P3D still surpasses the best competitor MARL by 1.1%.

Table 5: Performance comparisons with the state-of-the-art methods with RGB input on (a) UCF101 (3 splits)&HMDB51 (3 splits) and (b) ActivityNet.

(a) HMDB51 (H51) & UCF101 (U101)

| Method | Backbone | H51 | U101 |
|---|---|---|---|
| I3D (Carreira & Zisserman, 2017) | BN-Inception | 74.5 | 95.4 |
| ARTNet (Wang et al., 2018a) | BN-Inception | 70.9 | 94.3 |
| ResNeXt (Hara et al., 2018) | ResNeXt-101 | 70.2 | 94.5 |
| R(2+1)D (Tran et al., 2018) | ResNet-34 | 74.5 | 96.8 |
| S3D-G (Xie et al., 2018) | BN-Inception | 75.9 | 96.8 |
| STM (Jiang et al., 2019) | ResNet-50 | 72.2 | 96.2 |
| LGD-3D (Qiu et al., 2019) | ResNet-101 | 75.7 | 97.0 |
| **DG-P3D** | ResNet-50 | 77.4 | 97.3 |
|  | ResNet-101 | **78.8** | **97.8** |

(b) ActivityNet

| Method | Backbone | +K400 | Top-1 |
|---|---|---|---|
| TSN (Wang et al., 2018b) | BN-Inception |  | 72.9 |
| RRA (Zhu et al., 2018) | ResNet-152 |  | 78.8 |
| MARL (Wu et al., 2019) | ResNet-152 |  | 79.8 |
| TSN (Wang et al., 2018b) | BN-Inception | ✓ | 78.9 |
| MARL (Wu et al., 2019) | SEResNeXt152 | ✓ | 85.7 |
| **DG-P3D** | ResNet-50 |  | 76.5 |
|  | ResNet-50 | ✓ | 85.9 |
|  | ResNet-101 |  | 77.8 |
|  | ResNet-101 | ✓ | **86.8** |

Table 6: Performance comparisons with the state-of-the-art methods with RGB input on SS-V1 and SS-V2.

| Method | Backbone | Pre-train | SS-V1 | | SS-V2 | |
|---|---|---|---|---|---|---|
| | | | Top-1 | Top-5 | Top-1 | Top-5 |
| NL I3D+GCN (Wang & Gupta, 2018) | ResNet-50 | ImageNet+Kinetics | 46.1 | 76.8 | – | – |
| S3D (Xie et al., 2018) | BN-Inception | ImageNet | 47.3 | 78.1 | – | – |
| TSM (Lin et al., 2019) | ResNet-50 | ImageNet+Kinetics | 47.2 | 77.1 | 63.4 | 88.5 |
| bLVNet-TAM (Fan et al., 2019) | ResNet-50 | ImageNet | 48.4 | 78.8 | 61.7 | 88.1 |
| ABM-C-in Zhu et al. (2019) | ResNet-50 | ImageNet | 49.8 | – | 61.2 | – |
| I3D+RSTG (Nicolicioiu et al., 2019) | ResNet-50 | ImageNet+Kinetics | 49.2 | 78.8 | – | – |
| GST (Luo & Yuille, 2019) | ResNet-50 | ImageNet | 48.6 | 77.9 | 62.6 | 87.9 |
| STDFB (Martínez et al., 2019) | ResNet-50 | ImageNet | 50.1 | 79.5 | – | – |
| STM (Jiang et al., 2019) | ResNet-50 | ImageNet | 50.7 | 80.4 | 64.2 | 89.8 |
| **DG-P3D** | ResNet-50 | ImageNet | **51.8** | **81.2** | **64.5** | **90.0** |

Table 7: Comparisons with state-of-the-art methods on Kinetics-400 & Kinetics-600. The computational complexity is measured in GFLOPs × views and the views represent the number of clips sampled from the full video during inference. * In view that it is not that fair to directly compare irCSN pre-trained on IG65M (65M web videos) and other methods, here we report the performance of irCSN pre-trained on Sports1M.

| Method | Backbone | GFLOPs×views | Kinetics-400 (top-1/top-5) | | | Kinetics-600 (top-1/top-5) | | |
|---|---|---|---|---|---|---|---|---|
| | | | RGB | Flow | Fusion | RGB | Flow | Fusion |
| I3D | BN-Inception | 108×N/A | 72.1/90.3 | 65.3/86.2 | 75.7/92.0 | – | – | – |
| R(2+1)D | custom | 152×115 | 74.3/91.4 | 68.5/88.1 | 75.4/91.9 | – | – | – |
| S3D-G | BN-Inception | 66.4×N/A | 74.7/93.4 | 68.0/87.6 | 77.2/93.0 | – | – | – |
| NL I3D | ResNet-101 | 359×30 | 77.7/93.3 | – | – | – | – | – |
| LGD-3D | ResNet-101 | 195×N/A | 79.4/94.4 | 72.3/90.9 | 81.2/95.2 | 81.5/95.6 | 75.0/92.4 | 83.1/96.2 |
| X3D-XL | custom | 48.4×30 | 79.1/93.9 | – | – | 81.9/95.5 | – | – |
| irCSN* | custom | 96.7×30 | 79.0/93.5 | – | – | – | – | – |
| SlowFast | ResNet-50 | 65.7×30 | 77.0/92.6 | – | – | 79.9/94.5 | – | – |
| | ResNet-101 | 213×30 | 78.9/93.5 | – | – | 81.1/95.1 | – | – |
| | ResNet-101+NL | 234×30 | 79.8/93.9 | – | – | 81.8/95.1 | – | – |
| **DG-P3D** | ResNet-50 | 108×30 | 78.3/93.7 | 72.0/90.9 | 80.3/94.7 | 81.4/95.6 | 74.8/93.2 | 82.7/95.9 |
| | ResNet-101 | 218×30 | **80.4/94.7** | **73.2/91.2** | **82.5/96.0** | **82.6/95.9** | **76.7/93.4** | **84.3/96.6** |

Then, we turn to evaluate DG-P3D with optimization planning on four large-scale datasets, i.e., **SS-V1**, **SS-V2**, **Kinetics-400** and **Kinetics-600**. The top-1 and top-5 accuracies are reported on the four datasets. For Kinetics datasets, we additionally consider the flow modality for fair comparison with the baselines. The two-direction optical flow image is extracted by TV-L1 algorithm (Zach et al., 2007) in this paper. To reduce the time cost, the best optimization path found on the RGB modality of Kinetics-400 is utilized as the path for flow modality on Kinetics-400, and for both RGB and flow modalities on Kinetics-600. The results are shown in Table 6 and Table 7. Specifically, DG-P3D achieves the best performance with top-1 accuracy of 51.8% on SS-V1 and 64.5% on SS-V2. DG-P3D is superior to STM, which reports the best known results, by 1.1% and 0.3% respectively. On Kinetics-400, with only RGB input, DG-P3D achieves 80.4% top-1 accuracy, which makes the improvements over the recent 3D ConvNets irCSN (Tran et al., 2019), X3D-XL (Feichtenhofer, 2020), LGD-3D (Qiu et al., 2019), SlowFast (Feichtenhofer et al., 2019) by 1.4%, 1.3%, 1.0% and 0.6%, respectively. Such accuracy is even higher than that of the two-stream I3D (Carreira & Zisserman, 2017), R(2+1)D (Tran et al., 2018) and S3D-G (Xie et al., 2018). When fusing the prediction from both modalities, the accuracy of DG-P3D is further improved to 82.5%. The similar performance trends are also observed on Kinetics-600. The two-stream DG-P3D achieves 84.3% top-1 accuracy, which leads the performance by 1.2% against the best competitor of two-stream LGD-3D.

## 6 CONCLUSION

We have presented optimization planning which aims to automate the training scheme of 3D ConvNets. Particularly, a training process is decided by a sequence of training states, namely optimization path, plus the number of training epochs for each state. We specify the hyper-parameters in each state and the permutation of states determines the changes of hyper-parameters. Technically, we propose a dynamic programming method to seek the best optimization path in the pre-defined candidate transition graph and each state transition is stopped adaptively by estimating the knee point on the performance-epoch curve. Furthermore, we devise a new 3D ConvNets, i.e., DG-P3D, with a unique design of the dual-head classifier. The results on seven video benchmarks, which are different in terms of data scale, target categories and video duration, validate our proposal. Notably, DG-P3D with optimization planning obtains superior performances on all the seven datasets.

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

## A  APPENDIX

The appendix contains: 1) the collection of the performance-epoch curves; 2) the comparisons of 3D ConvNets architectures; 3) visualization of more optimization paths by optimization planning.

### A.1  CURVE COLLECTION

To evaluate the functions for knee point estimation, we pre-collect 162 performance-epoch curves from the training processes of different networks on different datasets. Particularly, P3D, G-P3D and DG-P3D networks are trained using hand-tuned strategy on six different datasets, i.e., HMDB51, UCF101, ActivityNet, SS-V1, SS-V2 and Kinetics-400. To obtain the curves with different settings, for each dataset, the network is firstly trained with 8-frame clips using 3-step learning rate strategy ([0.01, 0.001, 0.0001]), and then fine-tuned on 16-frame and 32-frame clips. Each step is trained for 50 epochs. Therefore, we collect $3 \times 6 \times 9 = 162$ curves in total.

### A.2  3D CONVNETS ARCHITECTURES

Table 8: Computational cost of different 3D ConvNets architecture.

| Network | I3D | R(2+1)D | LGD-3D | SlowFast | P3D | G-P3D | DG-P3D |
|---|---|---|---|---|---|---|---|
| **GFLOPs** | 108 | 152 | 195 | 234 | 196 | 203 | 218 |

Table 8 compares the computational cost of different network architectures. The number of floating-point operations (FLOPs) for one crop is given on each network. Overall, P3D network used in this paper requires similar computations with LGD-3D. The global context and dual-head classifier leads to 3.5% and 7.3% additional computations, respectively. Ultimately, the one-crop prediction of DG-P3D spends 218 GFLOPs, which is still lower than that of SlowFast network. The results basically indicate that our DG-P3D is potentially more economic and effective.

### A.3  VISUALIZATION OF MORE STRATEGIES

Figure 5 depicts the best optimization paths learnt by optimization planning on HMDB51, UCF101, ActivityNet, SS-V1, SS-V2 and Kinetics-400, respectively. We additionally show the best optimization paths on HMDB51 and UCF101 in Figure 5(g) and Figure 5(h), when taking Kinetics-400 for network pre-training. As expected, the training strategy predicted by optimization planning changes in response to different network initialization.

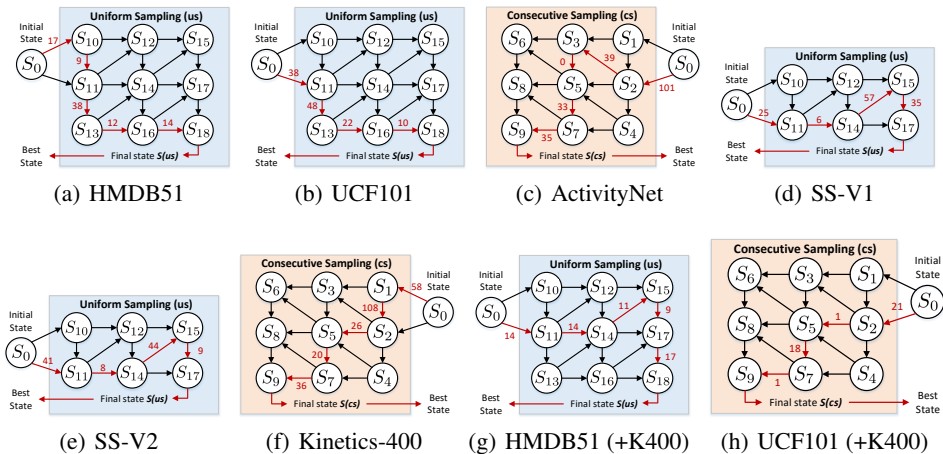

Figure 5: The best optimization path produced by the proposed optimization planning on (a) HMDB51, (b) UCF101, (c) ActivityNet, (d) SS-V1, (e) SS-V2, (f) Kinetics-400, (g) HMDB51 with Kinetics-400 pre-training and (h) UCF101 with Kinetics-400 pre-training. The red edge represents the state transition in the optimization path and the optimal number of training epochs is also given for each transition.

