# OpenReview forum: "Optimization Planning for 3D ConvNets"
_ICLR.cc/2021/Conference — Reject_

### Official Review · AnonReviewer1 · 2020-10-28
**Reviews**

**Rating:** 5
**Confidence:** 5

**Review:**

Summary:
  This paper proposed two things:
    1. hyper-parameter planning for training action recognition models.
    2. a new 3D net architecture for action recognition.

  The results show with the planning, authors can reduce training time significantly. and the proposed DG-P3D also is good for action recognition


Strong points:

  1. This paper explores a new approach for hyperparameter tuning to get a better model quickly.

  2. This paper performs extensive experiments to validate their proposed methods.

Weak points:

  1. It misses one very important reference on the same topic, "Multigrid Method for Efficiently Training Video Models" CVPR 2020. This paper also proposes a new way to efficiently train action recognition models.

  2. The way authors compared to others are mixed, making that it is a little bit difficult to compare the contribution of the paper.

    2.1 To be more fair compare to SOTA approaches in Table 4 and Table 5 with respect to the contributions of DG-P3D, the author should include the results without planning approach and use a similar training setting to the other SOTA approaches. Or apply the proposed optimization planning on those SOTA approaches to see the improvement. In this case, the advantages of each component will be more clear.

    2.2 When comparing to other approaches, authors should either include the number of frames used or computational load (like FLOPs) into the consideration, instead of performance only. E.g., for Kinetics400, slowfast-ResNet-50 (77.0% top-1 acc) only use 32 frames while the proposed models for Kinetics400 use 128 frames (If I understand correctly); thus, simply comparing the accuracy is unfair. On the other hand, If authors want to compare the performance only, authors might want to include the model with different backbones, e.g. bLVNet-TAM has models achieved 53.1% on SS-V1 and 65.2% on SS-V2 with RGB only.


Questions:

  1. If a node has multiple destinations, e.g. node S5 in Figure 1a, how do the author select S7 or S8? In my understanding, the proposed algorithm will train the model at S5 with the parameters at S7 and S8 independently and then select the better one. Is it correct?

  2. Followed by 1, when encountering multiple destinations, if all destination nodes need to be explored before making the transition, do the author takes those time into account as well in Table 4? I do not find those in Figure 4, either.

  3. What is the delay parameter T in practical? Is it a dataset-dependent parameter?

  4. What is the inference protocol used in the evaluation? The authors mentioned there are two common approaches but do not mention which one is used.

Others:

 1. When comparing to SOTA, I think authors miss "X3D: Expanding Architectures for Efficient Video Recognition" CVPR 2020.

After rebuttal:

The authors addressed some of my concerns, and the proposed optimization plan is interested.

1. I still think that the proposed architecture does not outperform SOTA architecture, like SlowFast.
  - First, in the rebuttal, the authors mention that with hand-crafted strategies, the proposed model is 0.4% better than SlowFast; nonetheless, the hand-crafted strategies train longer epochs than SlowFast. (SlowFast is trained with 196 epochs.) As authors propose new strategies, they know better how different strategies could affect.
  - Second, in the rebuttal, the authors improve I3D by 1.7% with optimization planning, which means, SlowFast might outperform the proposed model if involving optimization planning. Moreover, in the revised Table 7, the SlowFast does not have more FLOPs than the proposed network under the same backbone. Simply checking Table 8 might be confused.

2. About the optimization planning, it is still confused about how many epochs spent on the explored nodes. E.g. in Figure 4, for the Kinetics dataset, I do not understand why there is a black edge between S4 and S5 as S4 is not reached. (Authors noted that the black edges mean the explored strategies.) And why not there is no number on those explored edges? Another example, if we look at Table 3 and Figure 4 together, and again for the Kinetics dataset, the summation of those red numbers is 248, but this number is larger than any number in Table 3. As the authors mention that they included the epochs of the explored strategies in Table 3, that means they only spend 29 epochs in the exploration (if we treat Figure 4 for DG-P3D in Table 3.). For me, it seems too good to be true.

Thus, I keep my rating.

---

> ### Author Response · Authors · 2020-11-22
> **Responses to the comments of AnonReviewer1**
>
> Thank you very much for your great efforts and constructive comments. We appreciate the positive comments on 1) the planning can reduce training time significantly, 2) the proposed DG-P3D is good for action recognition, and 3) the paper performs extensive experiments.
>
> R1.1 Comparison with multigrid training
>
> Thanks for suggesting Multigrid Method as reference. Multigrid method proposes a new training strategy that cyclically changes spatial resolution and temporal duration of input clips for a more efficient optimization of 3D ConvNets. Our work contributes by studying not only training 3D ConvNets with multiple lengths of input clips, but also adaptively scheduling the change of input clip length through optimization planning. We will add the discussions with Multigrid method in the revision.
>
> R1.2.1 Include the results without planning; apply planning to other SOTA approach
>
> Thanks for your valuable comment. In Table 4 of the main paper, we summarized the comparisons between optimization planning and the best hand-tuned strategy on training DG-P3D across six different datasets. The best hand-tuned strategy can be considered as a well-tuned DG-P3D model without optimization planning. As indicated by the results, taking ResNet-50 as the backbone and training DG-P3D with the best hand-tuned strategy obtains 77.4%, which is higher than 77.0% of SlowFast on Kinetics-400 dataset. Furthermore, capitalizing on optimization planning for learning DG-P3D improves the accuracy from 77.4% to 78.3%.
> In addition, we conducted more experiments by exploiting the optimal path found with DG-P3D on Kinetics-400 as the path for I3D. Training I3D with such optimization path achieves the accuracy of 73.8% on Kinetics-400 with RGB input and leads to 1.7% performance improvement against the original I3D model.
> We will add these comparisons in our revision to ensure the advantages of our proposal are clear to readers.
>
> R1.2.2 FLOPs comparison
>
> Thanks for your valuable comment. We actually reported the FLOPS of single crop for different network architectures in Table 8 of Appendix. In general, the numbers of FLOPs of LGD-3D, SlowFast and DG-P3D are comparable. Please note that when using ResNet-101 as the backbone, the three 3D ConvNets all take 128-frame clips as inputs. In particular, both SlowFast and DG-P3D take multiple temporal stride into account, but they are different in the way that DG-P3D builds a dual-head classifier instead of the two-path structure in SlowFast. As such, DG-P3D costs less computations and is easier to implement. We will move the FLOPs comparisons to Table 7 for better clarification.
>
> R1.3 How to select the destination
>
> In our implementation, we actually choose the best preceding node rather than the destination node. For example, for node S5, we fine-tune the model at S3 and S2 with the hyper-parameters at node S5, and select the node, at which the fine-tuned model achieves better performance, as the preceding node of S5. To be clear, we will rephrase our statements to clarify this.
>
> R1.4 Time cost
>
> Yes, the determination of the best preceding relies on the requirement to explore all the possible transitions (from all possible preceding nodes), and the cost on those explorations is already taken into account in Table 4. In Figure 4, the black edges denote the transitions that have been explored but not selected in the final optimization path, while the red edges represent the states in the path. We will clarify this.
>
> R1.5 Delay parameter T
>
> Thanks for pointing this out. The delay parameter T is utilized to avoid non-trivial decision of the knee point and improve the stability of optimization. We simply fix T to 10 in all the experiments and did not even tune this parameter carefully. We will clarify the setting of delay parameter T in the revision.
>
> R1.6 Evaluation protocol
>
> As described in Section 5.2, we utilize the two inference strategies for planning optimization path and evaluating the learnt 3D ConvNets, respectively. In other words, the first strategy is only used when planning the optimization. Once the optimization path is fixed, we train 3D ConvNets with the path and evaluate the learnt 3D ConvNets by using the second strategy as in (Feichtenhofer et al. (2019)). To be clear, more details will be provided in the revised version.
>
> R1.7 The reference of X3D
>
> Thanks for suggesting X3D as reference. We will add the comparisons in our revision.
>
> \* We will submit the revision of our paper before November 24$^{th}$.

---

### Official Review · AnonReviewer3 · 2020-10-29
**Interesting paper about automate the optimization process of 3D ConvNets**

**Rating:** 6
**Confidence:** 1

**Review:**

- The paper proposed a Dual-head Global-contextual Pseudo-3D (DG-P3D) network and an automated optimization path to train 3D ConvNet for action recognition. The proposed method is evaluated on various of action recognition datasets, and achieved convincing results.

- Another contribution of the paper is its decreased cost. It mentioned that the time cost for grid optimization planning with 8 NVidia Titan V GPUs reduced the running time from 4057h to 288h on Kinetics.

- The optimization method is proposed for 3D ConvNet, However, the authors only benchmarked it with the new proposed DG-P3D network. It could be interesting to know what is the performance applying it to other basic 3D ConvNets. It is also interesting to know what is the performance of DG-P3D without this new optimization strategy.

---

> ### Author Response · Authors · 2020-11-22
> **Responses to the comments of AnonReviewer3**
>
> Thank you very much for your great efforts and constructive comments. We appreciate the positive comments on 1) the method achieved convincing results, and 2) its decreased cost.
>
> R3.1 Apply to other 3D ConvNets
>
> Thanks for the valuable comment. In the paper, we applied our optimization planning to P3D and G-P3D, which can be regarded as two substructures of DG-P3D. As shown in Table 3 of the main paper, optimization planning leads to the performance improvements over the best hand-tuned strategy by 1.4% and 0.9% on the two architectures, respectively. In view of this comment, we conducted additional experiments by exploiting the optimal path found with DG-P3D on Kinetics-400 as the path for I3D. Training I3D with such optimization path achieves the accuracy of 73.8% on Kinetics-400 with RGB input and leads to 1.7% performance improvement against the original I3D model. The results again demonstrate the effectiveness of our proposal on I3D network and also validate the generalizability of the learnt strategy across different networks. We will add the comparisons and discussions in the revision.
>
> R3.2 Performance of DG-P3D without optimization planning
>
> Thanks for your valuable comment. In Table 4 of the main paper, we summarized the comparisons between optimization planning and the best hand-tuned strategy on training DG-P3D across six different datasets. The best hand-tuned strategy can be considered as a well-tuned DG-P3D model without optimization planning. As indicated by the results, exploiting optimization planning can constantly lead to better performances than hand-tuned strategy across all the six datasets.
>
> \* We will submit the revision of our paper before November 24$^{th}$.

---

### Official Review · AnonReviewer2 · 2020-10-29
**Reviewer #2**

**Rating:** 6
**Confidence:** 4

**Review:**

Overview:
The paper proposes a novel way to automatically tune 3D ConvNet hyper-parameters (learning rate, input clip length, sampling way). This is achieved by decomposing the optimization path into several states and the state transition is triggered when the knee-point on the performance-epoch curve is met. Extensive experiments are conducted on popular video benchmarks and show that the optimization planning is effective to improve the accuracy and requires less time time compared to the hand-tuned procedure.


Strengths:

++ The paper provides a novel perspective of designing the hyper-parameters for 3D ConvNets. The automatic planning through the optimization path alleviate the tedious work of hand tuning and save a lot of times: Once the knee-point on the performance-epoch curve is achieved, we can easily transit to the next optimization state (with another set of hyper-parameters).


++ The experiments are thorough. The authors conduct experiments on all popular video recognition benchmarks. Also the experiment results corroborate the effectiveness of optimization planning.

Weaknesses:

-- The construction of transition graph has some strict principles that could be relaxed.
  (1) For scheduling learning rate, It only considers that "training starts from high learning rate". However, previous works (e.g. ICLR2017: https://arxiv.org/abs/1608.03983) have shown that cyclic cosine learning schedule might benefit training. Therefore, I don't see the necessity that we put this restriction.
  (2) For scheduling input length, this is not the first time that people do this. For example, Wang et. al. (CVPR2018, https://arxiv.org/abs/1711.07971) first train using 64 frames as input and then finetune on 128 frames. Also, Wu et. al. (CVPR2020, https://arxiv.org/abs/1912.00998) proposes another way of adjust the sampling strategy.


-- The comparison is mostly conducted on the newly proposed DG-P3D family architectures. It would be better if there is an additional row of results on the standard I3D/SlowFast network in Table 3. (this is somewhat minor and not required considering the time limit)


-- "82.5%" (with flow stream) is not the "new record" for Kinetics-400. ir-CSN-152 (ICCV2019, https://arxiv.org/abs/1904.02811) can achieve 82.6% on Kinetics-400 using **RGB only**. OmniSource (ECCV2020, https://arxiv.org/abs/2003.13042) improves it to 83.6% using additional data.




I have some additional questions:

-- In Figure 4, it shows that SS-V1/2 would prefer uniform sampling however Kinetics would prefer consecutive sampling. However, this seems somewhat counter-intuitive to me. We know that Kinetics have videos with static scenes or slow motion while SS-V1/2 have some videos with more intensive motion. Does it make more sense to use dense frames (consecutive) to recognize intensive motion?

---

> ### Author Response · Authors · 2020-11-22
> **Responses to the comments of AnonReviewer2**
>
> Thank you very much for your great efforts and constructive comments. We appreciate the positive comments on 1) the paper provides a novel perspective, 2) the automatic planning alleviates the tedious work of hand tuning, and 3) the experiments are thorough.
>
> R2.1 Design principles
>
> R2.1.1 Scheduling learning rate
>
> Thanks for the point. Yes, some very specific learning rate strategies, e.g., schedules with restart or warmup, show that increasing the learning rate properly may benefit training. Nevertheless, there is still no clear principle of when to increase the learning rate, and thus it is very difficult to automate these schedules. In the works of adaptively changing the learning rate for 2D ConvNets training (Ge et al., 2019; Lang et al., 2019; Yaida, 2019), such cyclic schedules are also not taken into account. As a result, we only consider the schedule of decreasing learning rate with multiple steps. We will add the explanations to clarify this.
>
> R2.1.2 Scheduling input length
>
> Yes, the recent works, e.g., Wang et al., CVPR 2018 and Qiu et al., CVPR2019, first train 3D ConvNets with short input clips and then fine-tune the network with lengthy clips. Such design is to balance training efficiency and long-range temporal modeling. Ours derives from the similar spirit but unlike the existing works which manually determine the number of training epochs for each clip length, our research contributes by automating the transition between different lengths of input clips. We will clarify this.
>
> R2.2 Results on other 3D CNNs
>
> Thanks for your valuable comment. We conducted additional experiments by exploiting the optimal path found with DG-P3D on Kinetics-400 as the path for I3D. Training I3D with such optimization path achieves the accuracy of 73.8% on Kinetics-400 with RGB input and leads to 1.7% performance improvement against the original I3D model. The results again demonstrate the effectiveness of our proposal on I3D network and also validate the generalizability of the learnt strategy across different networks. We will add the comparisons and discussions in the revision.
>
> R2.3 “82.5%” is not the new record for Kinetics-400
>
> Thanks for pointing this out. Yes, ir-CSN-152 did achieve the higher accuracy (82.6%) on Kinetics-400 and OmniSource further improves ir-CSN-152. However, ir-CSN-152 is pre-trained on a web video dataset (IG65M), which consists of 65M web videos and is multiple orders of magnitude larger than Kinetics-400, and OmniSource capitalizes on even more data. As such, it is not that fair to directly compare ours and the two works. Please also note that without pre-training on IG65M or pre-training on Sports1M instead, ir-CSN-152 only achieves 76.8% or 79.0% on Kinetics-400, respectively, which is lower than 80.4% of DG-P3D with ResNet-101 as the backbone. Considering this comment, we will add the discussions and tune down our claim in the revision.
>
> R2.4 Sampling strategy of Kinetics-400 and SS-V1/2
>
> We appreciate your insightful comment on the observations of the preferred sampling strategies for different kinds of datasets. In general, the most special on uniform sampling is to capture the completeness of a video with only a small number of sampled frames. In contrast, consecutive sampling emphasizes the continuity in a video but may only focus on a part of the video content. Such characteristics lead to different preferences on sampling strategy for different datasets. The SS-V1/V2 datasets consist of fine-grained interactions and the differentiation between these interactions relies more on the completeness of an action. For example, it is almost impossible to distinguish the videos of the category “Pushing something so that it falls off the table” from those of “Pushing something so that it almost falls off but doesn't” or “Pushing something so that it slightly moves”, if only based on part of the video content. In other words, uniform sampling offers the completeness of a video and benefits the recognition on SS-V1/V2. In contrast, the videos in Kinetics-400 are usually with static scenes or slow motion. Hence, the completeness may not be essential in this case, but consecutive sampling encodes the continuous changes across frames and thus captures the spatio-temporal relation better. We will add the discussions in the revision.
>
> \* We will submit the revision of our paper before November 24$^{th}$.

---

### Official Review · AnonReviewer4 · 2020-10-30
**This paper proposes an optimization planning strategy to systematically train a 3D network. It shows promising results on multiple datasets for the tasks of action recognition.**

**Rating:** 7
**Confidence:** 5

**Review:**

- The overall quality is good. It addresses the training issue of 3D convolutional network in a novel perspective and shows promising results. The presentation is clear except for some grammar mistakes.

Detailed Comments:

- In Sec. 3.2, how is the transition performance evaluated is not clear. What is the cost / gain from i to j? Is it the same for all states?

- Ablation experiments should be provided to show the performance difference using different paths. Is the adopted path actually the optimal path? And how about if some intermediate states are skipped?

- The proposed optimization planning doesn't seem to work specifically for the 3D convnets and for the action recognition task. As long as we need to train a network in stages with multiple hyper-parameters, this should apply theoretically.

- Abstract: 'follows' --> 'is followed'

---

> ### Author Response · Authors · 2020-11-22
> **Responses to the comments of AnonReviewer4**
>
> Thank you very much for your great efforts and constructive comments. We appreciate the positive comments on 1) the idea is in a novel perspective, 2) the results are promising, and 3) the presentation is clear.
>
> R4.1 How is the transition performance evaluated?
>
> When executing the transfer from the state $S_i$ to the state $S_j$, we fine-tune the 3D ConvNets at the state $S_i$ by using the hyper-parameters at the state $S_j$. We then evaluate such fine-tuned model on the validation set to measure the priority of this transition, i.e., $\mathcal{V}(S_i \to S_j)$. We choose the state $S_i$, which achieves the highest priority of transition to the state $S_j$, as the preceding state of $S_j$. We will rephrase the statements to better clarify this.
>
> R4.2 Performance difference using different paths; is the adopted path actually the optimal path? how about if some intermediate states are skipped?
>
> Thanks for your valuable suggestions. Table 3 and Table 4 in the main paper summarize the comparisons between hand-tuned strategies and our optimization path. The results consistently demonstrate the advantage of our optimization planning across different 3D ConvNets and datasets. In view of this comment, we experimented with some variant paths on UCF101, which are built by either inserting an additional state or skipping an intermediate state in our adopted path. For fair comparisons, the numbers of epochs in these variant paths are re-determined by the algorithm in Section 3.3. The results indicate that inserting and skipping one state result in an accuracy decrease of 0.2%$\sim$1.7% and 0.3%$\sim$1.5%, respectively. We will add the comparisons and discussions in the revision.
>
> R4.3 Apply the proposed optimization planning to other networks
>
> We appreciate your insightful comment of applying the proposed optimization planning to train other types of networks for different tasks. Yes, our optimization planning is readily applicable to train other networks. Our focus in this paper is on the problem of network training for 3D ConvNets, from three standpoints: 1) the training scheme of 3D ConvNets contains several complex choices; 2) the current works on 3D ConvNets usually capitalize on specific training strategies; and 3) most of the existing training strategies of 3D ConvNets can be formulated as one candidate path in the transition graph. The suggestion on experimenting our optimization planning on other networks is much appreciated, which would be one of our future works. We will add these explanations.
>
> R4.4 Grammar mistakes
>
> Thanks. We will carefully go through the paper and fix the grammar mistakes in our revision.
>
> \* We will submit the revision of our paper before November 24$^{th}$.

---

### Author Response · Authors · 2020-11-25
**Paper Revision**

Dear Reviewers,

The authors would like to show our appreciation to the referees of the submitted manuscript for the valuable and constructive comments. According to the comments, we have made corresponding revisions and updated our revised version. Thank you very much.

---

### Decision · Program_Chairs · 2021-01-07
**Final Decision**

**Decision:**

Reject

**Comment:**

The reviewers appreciate the idea of hyperparameter planning and the thorough experimentation. Some concerns remain regarding the comparison between this method and SlowFast that require to be addressed. Also, the scope of the paper that targets hyperparameter optimization networks for action recognition specifically, may be too narrow for an ICLR audience.